# Evaluation of the Satisfaction of Patients Seen in the Dermatology Department of a Spanish Tertiary Hospital

**DOI:** 10.3390/healthcare10081560

**Published:** 2022-08-18

**Authors:** Camino Prada-García, José Alberto Benítez-Andrades

**Affiliations:** 1Servicio de Dermatología, Complejo Asistencial Universitario de León, 24008 León, Spain; 2SALBIS Research Group, Department of Electric, Systems and Automatics Engineering, Campus of Vegazana s/n, University of León, 24071 León, Spain

**Keywords:** patient satisfaction, perceived quality, dermatology, SERVQHOS

## Abstract

Patient satisfaction is of great importance and is a key factor in the quality of care. The most commonly used measure of this factor is satisfaction surveys. This study used the modified SERVQHOS healthcare quality survey model, which adapts the SERVQUAL survey. The main objective was to determine the degree of satisfaction of patients seen in the outpatient department of the Dermatology Service, as well as to describe and detect those aspects that could be improved to offer better quality care. A total of 250 patients responded to the survey. The mean Likert scale score for the 19 items on the perceived quality of care was 4.17 ± 0.796 points. Up to 92.8% were satisfied or very satisfied with the care received. All items were statistically correlated with overall satisfaction (*p* < 0.001). In the multivariate study, the variables with predictive capacity in relation to overall satisfaction (*p* < 0.05) were “the technology of the medical equipment”; “the directions to the consultation”; “the confidence that the staff transmits”; “the state of the consultation”; and “the interest of the staff in solving problems”. Satisfaction was significantly higher in men (*p* < 0.05), with a level of education up to primary school (*p* < 0.05) and no work activity (*p* < 0.001). The final mean score in the degree of perceived satisfaction was very high, indicating that the expectations of the patients were exceeded, and showing that satisfaction is closely linked to the qualities and skills of the staff in their relationship with the patient.

## 1. Introduction

Quality of service is difficult to define, as objective and subjective elements relating to user satisfaction are involved [1]. This refers to the totality of characteristics and attributes of a service that affect its ability to satisfy a given need. However, it is increasingly expressed in terms of the consumers’ expectations of the service to be provided, compared to their perceptions of the experience [2].

From a management point of view, quality can be defined as the set of properties and characteristics of a product or service that make it capable of satisfying expressed or implicit needs, which implies detecting these needs, comparing them with a reference model and meeting these needs and customer expectations. Users’ opinions about the services received are key in defining quality and their assessment is essential in order to provide correct health care [3].

With regard to the quality of care, numerous definitions have been made over the years, valid depending on the context in which they are applied. Thus, in 1980, Avedis Donabedian defined it as the model of care expected to maximise the patient’s level of well-being, after taking into account the balance of expected benefits and losses in all phases of the care process, while in 2000, the WHO considered it as the level of realisation of intrinsic objectives to improve health by health systems, and of responsiveness to the legitimate expectations of the population [4].

Healthcare quality has two distinct facets, technical quality and functional quality. Technical quality refers to the accuracy of medical diagnoses and procedures and is generally understood by the professional community but not by patients. Patients essentially perceive functional quality as the way in which services are delivered [2]. Thus, technical quality is about what patients obtain; it refers to the outcome of the service provided and is determined by factors such as days of hospital stay, admission rate or morbidity and mortality. On the other hand, functional quality is about how they obtain the service provided and is defined by factors such as the professionals’ attitudes towards the patients, the facilities or the hospital’s equipment [5,6].

It is a complex concept that depends on patient characteristics and changes with age, gender, educational and socioeconomic level and changes in health status [7]. Moreover, these characteristics influence the attitude of professionals, which in turn has an impact on patient satisfaction [8]. There is theoretical and empirical evidence that the quality of a service sustains satisfaction and that they are mobilised together, but they are not equivalent concepts, they are interdependent [9].

Quality of service largely determines patient satisfaction. It is a concept that can be popularly defined as conformity with the patient’s expectations and serves as a benchmark in the evaluation of medical care [10].

In recent decades, the quality of care in hospitals has become a primary objective, and in the 1990s the so-called continuous quality improvement process was introduced [6]. The aim of continuous improvement in the quality of care is to care for and promote perceived quality, encouraging patients to exceed their expectations. This improvement process must be in constant review and improvement of the quality of care offered [11].

Measuring service quality is the first step toward quality improvement and quality management in health systems [12]. Research on service quality and consumer satisfaction evaluation studies are fundamental to the continuous improvement of healthcare organisations and is a concept that patients highly value, as well as healthcare professionals and organisations alike [13].

Measuring quality in healthcare is necessary to improve patient satisfaction [14]. Donabedian formulated that the strategy for evaluating the quality of a care service requires an analysis of the structure (resources), the processes or methods used and the results. However, for a correct evaluation of quality, it is necessary to use indicators, which implies the prior elaboration of quality criteria, defined as those desirable or undesirable conditions that certain relevant aspects of health care must fulfil. The indicator is a quantitative measure expressed as a percentage of the degree of compliance with the criteria, and it helps to detect points for improvement or change [15].

Quality as a concept has multiple dimensions, one of which is patient satisfaction. It is also considered an important outcome indicator for assessing the quality of care provided [16].

The first to state that patient satisfaction is first and foremost a measure of the outcome of the interaction between the health professional and the patient was Koos and Donabedian [17].

The concept of patient satisfaction has evolved in recent years, moving away from the traditional paternalistic relationship between doctor and patient to a scenario in which the user’s opinion, as well as the factors that increase their safety and satisfaction, play a major role [18]. As a result, the patient has become the main driver and focus of health services, and satisfaction is defined as the extent to which health care and the resulting health status meet the user’s expectations. The patient is, therefore, considered to be satisfied to the extent that the services provided meet or exceed his or her expectations [11,19].

Interest in patient satisfaction has increased in recent years. It is considered a useful instrument for assessing the outcome of the care process, providing important information about the quality received, and can be incorporated in order to improve it, so it is important to know the patients’ view. This satisfaction identifies different dimensions of care, such as technical aspects, communicative processes and comfort of care [20]. Its complexity lies in the fact that it is related to numerous factors, such as lifestyle, previous experiences, expectations and values [21]. Knowing patient satisfaction can predict, among other things, therapeutic adherence and return to the same health care centre in the event of another episode. In this way, patient dissatisfaction would lead to less confidence in the health care centre and, consequently, to a loss in the number of patients who “return” to it, as well as to a deterioration of the image that would damage the entire health care organisation [22,23].

The results obtained when analysing whether the patient’s age, level of education or income determine their level of satisfaction are contradictory. However, there is a slight tendency to find greater satisfaction in older patients or those with medium and higher incomes. In addition, it has been reported that women rate medical care more positively. The confidence that the doctor inspires in the patient is a good indicator of the degree of satisfaction, which varies according to certain characteristics of the doctor–patient interaction, such as the length of the consultation or communication skills [17]. The study of the relationship between sociodemographic factors and satisfaction will allow a more realistic view to be obtained by patient subgroups, to adapt those aspects identified as deficient to the first ones [24].

It is impossible to adequately describe quality without patient feedback, as it provides information about the success or failure of the health system in meeting patients’ expectations and has become the central focus of public health services [13,21]. Thus, patient satisfaction is one of the indicators to measure the effectiveness of healthcare management and is also considered the main indicator of the quality of care [25]. The needs of the patient are the pivotal point around which care services must be articulated and form the basis for the organisation of the services and of the hospitals themselves [11,26].

Patient satisfaction is a desirable element of health care outcome, which is why this measure has been extended to both primary care and hospital care [27]. Several tools have been developed to measure patient satisfaction with regard to their perceptions and expectations. These tools vary in their definitions, content and measurement, ranging from model surveys with open-ended, general questions to questionnaires with structured, weighted questions. Their main objectives are to find out the characteristics of the care offered by the hospital and to find out what service patients want [28,29].

The study of satisfaction provides information both to health professionals and health system managers and administrators on perceived quality, allowing it to be integrated as a measure for improving the quality of care [19]. It also makes it possible to know the sociodemographic and health care characteristics that influence satisfaction, in order to be able to improve and provide a better quality service [30].

The SERVQUAL scale, developed by Parsuraman et al. in 1988 [31], is one of the most widely used scales for measuring patient satisfaction and service quality. This scale is designed to assess patients’ expectations and perceptions of service quality using 22 items representing five dimensions, using a seven-point Likert scale ranging from “strongly agree” to “strongly disagree” [2,10,12,32,33,34,35,36,37]:-Tangibility (four items): refers to physical facilities, equipment, materials and personnel appearance;-Reliability (five items): ability to perform the promised service reliably and accurately;-Responsiveness (four items): willingness to help customers and provide prompt service;-Assurance (four items): competence, courtesy, credibility and security;-Empathy (five items): individualised care and attention given to patients by health care staff.

The measurement of quality through this questionnaire is based on the “disconfirmation paradigm”, which expresses the difference between what the customer expects (expectations) and what he/she perceives (perceptions). In this way, satisfaction will be all the greater, the more these expectations are exceeded [11,38].

The SERVQHOS questionnaire [26], an adaptation of the SERVQUAL survey [31], is designed to measure patient satisfaction after hospitalisation in Spanish-speaking countries. This evaluation model consists of a first part with 20 items on different care factors, with Likert-type responses ranging from 1 (much worse than expected) to 5 (much better than expected), and distinguishing between aspects related to subjective quality and organisational and facility-related aspects (objective quality); a second section of additional questions, including one on overall satisfaction; a third block in which sociodemographic data are collected; and finally, a section in which there is a free space for suggestions or comments [19]. It is a questionnaire based on the “disconfirmation paradigm”, which measures satisfaction as the difference between what the patient expects and what he or she perceives [28]. Its advantages include a small number of items and ease and speed of response. However, some of its items need to be adapted in contexts other than the general hospital environment and in specific services. Moreover, it is an instrument which, when used periodically, offers the opportunity to monitor the quality perceived by patients and detect potential areas for improvement, enabling management and staff to make decisions. This scale has demonstrated its ability to discriminate between satisfied and dissatisfied patients and a sufficient interrelation with an overall measure of patient satisfaction [39].

There are few studies at the international level that analyse the satisfaction of patients seen in dermatology outpatient clinics. Thus, we decided to carry out a study in the Complejo Asistencial Universitario de León (CAULE) to find out patients’ opinions, measure their satisfaction and detect areas for improvement to offer efficient, quality care.

The main objective of this study was to determine the degree of satisfaction of patients seen in the outpatient department of the CAULE Dermatology Department, according to the modified SERVQHOS questionnaire, as well as to describe and detect those aspects that could be improved in order to be able to offer better quality care.

The specific objectives were:-To analyse the items best and worst rated by patients in the SERVQHOS questionnaire;-To identify the aspects that generate greater and lesser patient satisfaction;-To analyse the possible association between the level of satisfaction and the patients’ sociodemographic variables;-To analyse the possible association between the level of satisfaction and the additional questions corresponding to the second part of the questionnaire;-To analyse the possible association between the sociodemographic variables and each aspect of care in the first part of the questionnaire;-To describe the comments and suggestions made by patients in the SERVQHOS questionnaire.

## 2. Materials and Methods

A cross-sectional observational study was conducted, including all patients who completed the satisfaction questionnaire (SERVQHOS model) between 1 November 2020 and 31 March 2021. The study population consisted of patients attending the outpatient clinics of the CAULE Dermatology Department. The sample consisted of 250 surveys completed by patients and handed out at the exit of the outpatient dermatology department of the CAULE. All the participants in this study met the inclusion criteria below and voluntarily signed the informed consent form giving their acceptance, with the opportunity to decline participation at any time during the study.

The inclusion criteria were as follows:
Patients over 18 years of age who attended the outpatient clinic of the Dermatology Department of the CAULE and who voluntarily signed the informed consent form to participate in the study;Patients under 18 years of age who attended outpatient consultations at the CAULE Dermatology Department and whose parents or guardians signed the informed consent form to participate in the study. These patients responded with the help of an adult and the items that they did not understand correctly were explained to them.

The exclusion criteria were as follows:
Patients with apparent and/or diagnosed mental disability;Patients with language impairment;Patients with altered state of consciousness (Glasgow score less than 15/15);Patients who could not read and write;Very old patients with an inability (cognitive impairment) to complete the questionnaire and the informed consent form.

Data collection was carried out through a modification of the SERVQHOS questionnaire that was adapted to the consultations. The participants were informed of their participation in collecting information to assess the quality of care, and the survey was anonymous and voluntary. For this purpose, patients could access the questionnaire via a link or a QR code. This survey was given to the patients or to the person accompanying them if the patient was mentally or physically disabled after the end of the consultation.

The survey consisted of a first block with 19 items relating to health care; ten questions assessed subjective quality (interest in solving problems, interest in keeping promises, speed of response, willingness to help, trust and safety, friendliness of staff, preparation of staff, personalised treatment, understanding of needs and interest in nursing) and nine questions assessed objective quality (technology, appearance of staff, signposting to the consultation, punctuality of consultations, comfort, information given by the doctor, waiting time, ease of reaching the care centre and information for relatives). These items were scored on a Likert scale from 1 (the care received was much worse than expected) to 5 (the care received was much better than expected). In this way, the highest scores referred to those aspects that patients rated positively. The second block consisted of seven additional questions, one referring to the overall satisfaction level. The third block was related to sociodemographic information (age, sex, level of studies, employment status) and the last block presented an open-ended question, in which the patient could make suggestions or comments for improvement regarding the care received in the consultation.

The database generated was completely anonymous, and no variables relating to the patient’s personal identification were collected. The patients who chose to participate read and accepted the informed consent form before starting the survey. All data were treated with absolute confidentiality, being used only for the purposes described as objectives of the present study.

At the meeting of the Clinical Research Ethics Committee of León on 25 November 2020 (Registration Number: 20189), it was unanimously agreed, as the methodological and ethical aspects of the study were considered correct, to approve the project.

The data obtained were recorded in an Excel spreadsheet for the study and statistical treatment of the variables. Subsequently, they were analysed with the IBM SPSS Statistics v. 25 statistical programme. Categorical variables were expressed as frequency and percentage, while measures of central tendency and dispersion (means and standard deviation) were calculated for quantitative variables. Pearson’s correlation coefficient was used to analyse the joint variation of overall satisfaction with the different items studied. Subsequently, a multivariate analysis was performed using logistic regression to identify the variables that determined patients’ overall satisfaction. In addition, an analytical study was carried out to obtain the possible associations between the sociodemographic variables of the patients and the level of satisfaction achieved, finding the existence or not of statistically significant differences, as well as between the variables of the second part of the questionnaire and the degree of overall satisfaction, and between the sociodemographic variables and each aspect of care in the first block of the questionnaire. The association between qualitative variables was performed using Pearson’s Chi-square test, which allowed us to test the hypothesis that the variables were independent. In cases where more than 20% of the expected frequencies were less than 5, Fischer’s exact test was used for 2 × 2 tables or the likelihood ratio for tables larger than 2 × 2. The significance level was set at *p* < 0.05 and the confidence intervals at 95%.

## 3. Results

### 3.1. Analysis of Sociodemographic Characteristics

Of the 250 patients participating in the survey, 145 were female (58%) and 105 male (42%). The patients ranged in age from 10 to 95 years, with a mean age of 45.4 years. When analysing the age data by ranges, the largest age group was 45–54 years (20% of patients); followed by 35–44 and 15–24 years (both 17.6% of patients); and 55–64 years (16.0% of patients). With regard to marital status, most of the patients were single (44.4%, *n* = 111) or married (43.6%, *n* = 109), with a very low percentage of those who were widowed (8.0%, *n* = 20) or separated (4.0%, *n* = 10). In terms of educational level, 98 patients (39.2%) had completed university studies; 72 patients had finished high school (28.8%); 69 patients (27.6%) had a primary education; and only 11 patients (4.4%) had no education at all. About employment status, most patients were working (53.6%, *n* = 134); 48 patients were students (19.2%); 47 were retired (18.8%); and only 21 patients were unemployed (8.4%).

### 3.2. Analysis of Variables Relating to Quality of Care

The analysis of the means of the variables assessed in the first part of the questionnaire showed that the majority of patients were satisfied in the dimensions explored, with a score above 4 in all of them except for the item “the technology of the medical equipment for diagnosis and treatment”, which was 3.68 ± 1.072 (on a scale of 1 to 5 where 1 is much worse than expected and 5 is much better than expected), meaning that all the items were rated at or above expectations.

The aspects that patients were most satisfied with were “the friendliness (politeness) of the staff in their dealings with the patient” (4.44 ± 0.796); followed by “the appearance (cleanliness and uniform) of the staff” (4.31 ± 0.854); and “the preparation (training) of staff to do their job” (4.28 ± 0.953). The most valued aspects were “the technology of medical equipment for diagnosis and treatment” (3.68 ± 1.072) and “the information that the dermatologist gives to the relatives” (4.00 ± 1.012), although it should be borne in mind that in this last item a larger sample size was assessed (*n* = 83), since, due to the pandemic, no relatives were allowed access to the consultation except in cases of need (minors or dependents), meaning that 66.8% of the patients attended alone (Table 1).

The average Likert scale score for the 19 items on the perceived quality of care was 4.17 ± 0.796 points.

### 3.3. Analysis of Additional Questions

With regard to the second part of the questionnaire, when asked about the overall level of satisfaction with the health care received during the dermatology consultation, 48.8% were very satisfied (122 patients) and 44.0% were satisfied (110 patients), while 6.4% (16 patients) and 0.8% (two patients) were dissatisfied or not satisfied, respectively, with the care received at the dermatology clinic. A total of 88.0% of the respondents (220 patients) answered that they would recommend the service received at the dermatology clinic to others without hesitation, while 11.2% would have doubts (28 patients) and only 0.8% said they would never recommend it (two patients). Regarding the opinion on the consultation time, 87.2% of the respondents considered that they had spent the necessary time (218 patients), while 10.4% responded that the time was less than necessary (26 patients), and for 2.4% the consultation time was longer than necessary (six patients). When analysing the dichotomous questions in the second block, 96.8% of the respondents (242 patients) answered that no tests or interventions had been performed in the consultation without their permission; 43.6% (141 patients) said they knew the name of the doctor who attended them in the consultation; while 86.4% (216 patients) said they did not know the name of the nurse who had attended them. In addition, 82.8% (207 patients) felt that they had received sufficient information about what was wrong with them.

### 3.4. Study of the Association between Variables

Correlation analysis

A correlation analysis between the different variables in the first block of the questionnaire (19 items) and the level of overall satisfaction with the health care received in the dermatology clinic showed that all of them were related to overall satisfaction (*p* < 0.05).

Logistic regression analysis

A multivariate logistic regression analysis was carried out to predict the degree of overall satisfaction in relation to the variables studied in the first section of the questionnaire, in which only five of them were statistically significant (*p* < 0.05) (Table 2).

Analysis of the association between the level of satisfaction and sociodemographic variables.

Some of the patients’ sociodemographic variables (sex, age, marital status, completed studies and employment status) were grouped to analyse their association with perceived overall satisfaction. For this purpose, three age ranges were differentiated (≤45 years, 46 to 65 years and ≥66 years); completed studies were divided into higher studies (baccalaureate, university) and even primary studies (no studies, primary); marital status into married and unmarried (single, widowed, separated); sex into male and female; and current employment status was differentiated between those who were working and those who were not (unemployed, students, retired). For the level of satisfaction, two categories were established: satisfied (very satisfied and satisfied) and not satisfied (not very satisfied and not at all satisfied). Statistically significant differences were found in the variables sex (*p* = 0.024); completed studies (*p* = 0.049); and employment status (*p* = 0.009), and it was observed that the people who were most satisfied were those who were male, with a level of studies up to primary school and without a job (Table 3).

Analysis of the association between the level of satisfaction and the additional questions corresponding to the second part of the questionnaire.

We analysed the possible association between the level of satisfaction recoded into satisfied (very satisfied and satisfied) and not satisfied (not very satisfied and not at all satisfied) with the variables corresponding to the additional questions in the second part of the questionnaire, finding statistically significant differences in the variables recommendation of the service received (*p* < 0.001); consultation time (*p* < 0.001); and sufficient information about what was wrong (*p* < 0.001). It was observed that the patients who were most satisfied were those who recommended the dermatology service without hesitation (99.5%); those who considered that the consultation time was as long as necessary (95.9%) or longer than necessary (100.0%); and those who received sufficient information about what was wrong with them (99.0%) (Table 4).

Analysis of the association between sociodemographic variables and each aspect of care in the first block.

Finally, the sociodemographic variables of the patients were assessed, related to each aspect of care in the first block (19 items), both recoded, to group the data and response alternatives (Table 5).

Comments from the open question in the questionnaire

An analysis of the comments made by the patients in the final section of the questionnaire was carried out and the following suggestions were mainly made:-Reduced waiting time for care and test results;-Closer and continuous follow-up with more frequent check-ups;-More information and attention time from doctors;-Refurbishment of the facilities and furnishings of the consultation and waiting room;-Treatment of benign/aesthetic lesions;-Increased research;-Telematic communication with doctors;-Fewer changes of doctor and knowing the doctor’s identity;-Increase in the salaries of health personnel;-Improved coordination between outpatient clinic/hospital;-Greater punctuality;-Closer attention.

## 4. Discussion

In this study, the SERVQHOS questionnaire [39] was used to measure perceived quality using a Likert scale, assessing expectations and perceptions of the service received to evaluate satisfaction as defined by the “disconfirmation paradigm”. Surveys are a simple tool for assessing perceived quality. They make it possible to find out how patients evaluate the service received and, thus, establish feedback to modify all aspects that could be improved and subsequently, evaluate the impact of these actions aimed at correcting errors in the service.

The literature review found only five published studies on satisfaction in dermatology services, but none of them used the SERVQHOS questionnaire [40,41,42,43,44].

Regarding the sociodemographic profile, the results show that the average age of patients seen in dermatology outpatient clinics is 45.42 years, with a very wide age range, between 10 and 95. A total of 67.6% of the respondents were aged between 10 and 54 years, indicating that the majority were young adults, mainly female (58.0%), with a marital status of single (44.4%) or married (43.6%) as the most frequent choices. Most of the patients (68%) had completed higher education (high school or university) and, as for employment status, more than half of the patients (53.6%) were working.

The 19 items on the different aspects of health care were assessed using five response alternatives. The scores were high, showing that most patients were satisfied in all the dimensions explored. These results are consistent with other studies reviewed. The aspects of health care with the highest scores corresponded to “the friendliness of the staff in their dealings with the patient”; “the appearance of the staff”; “the preparation of the staff for their work”; “the personalised treatment received”; “the confidence that the staff transmits”; “the waiting time to be seen”; and “the punctuality of the consultation”. In contrast, the lowest-rated aspects were “the technology of the medical equipment”; “the information the doctor gives to relatives”; “the state of the consultation room”; “the speed with which you get what you need or ask for”; “the ease of getting to the hospital or outpatient clinic”; and “the directions to the consultation room”. In general terms, therefore, patients are more satisfied with the aspects related to professional competence, the human component or the treatment they receive from the healthcare staff (subjective quality), while the worst evaluations fall on the more tangible aspects related to the structure of the centre, logistics, accessibility or the technology of the equipment (objective quality), aspects that could be modified, with the consequent improvement in the quality received by the patient. These results have also been found in other published studies [18,24,26,45,46,47,48] and confirm the importance of a professional but close relationship between all those involved in the patient care process. However, the negative results are more interesting than the achievements in patient satisfaction, because of the possibilities for improvement that may arise.

The mean score on the Likert scale for the 19 items on the perceived quality of care was 4.17 out of 5, which shows that patients rated their stay in the dermatology clinic very positively and can, therefore, say that they consider the service to be “better than expected”.

The results obtained in the second block of questions of the questionnaire (benchmarks) were highly satisfactory:

92.8% of patients were satisfied or very satisfied with the health care they received. These results are similar to previous studies, which show that patients’ perception of the service received is higher than their expectations [1,18,26,28,45,46]. These high figures are important, as patient satisfaction has been related to relevant aspects such as adherence to treatment, consultation and professional recommendations, and has even been related to an improvement in health status [24]. Furthermore, 88% of respondents would recommend the service received at the dermatology clinic to others without hesitation and only two patients (0.8%) said they would never recommend it, figures also similar to those of other studies [28,46];96.8% of the patients responded that they had always been asked for their permission for tests or interventions, so that, in almost all cases, the patient has the final decision on aspects related to his or her health. Apart from that, legally, it is a prerequisite for any intervention to be carried out. Regardless of the legal aspects, the lack of consent is often an aspect that deteriorates the doctor–patient relationship;The interpersonal relationship between the patient and the healthcare team was also assessed. They were asked whether they knew the name of the professionals who attended them. In this section, the results were not positive. Up to 56.4% of the patients stated that they did not know the doctor’s name, and 86.4% did not know the nurse’s name. There is a greater lack of knowledge of the nurse’s name, an aspect that coincides with what has been published in other studies [24,26,28,45,46]. This may be due to the fact that, although all health staff introduce themselves the first time they contact the patient, it is easier to remember the name of the doctor, as it is not always the same nurse who attends the patient, making it more difficult for the patient to remember her name and making identification more difficult. Improving this aspect could lead to a greater appreciation of the relationship of trust between the patient and the health staff. In this way, introducing oneself by giving one’s name may be useful to reduce patient mistrust in the consultation room and improve therapeutic compliance, so that, although the data in this study are consistent with those published in other studies, this is an important aspect that should be improved. In addition, the incorporation of a “referring nurse” could solve this problem and improve patient satisfaction;Regarding the information received, 82.8% considered that they had received sufficient information about what was happening to them. However, “the information given by the doctor to the relatives” was one of the worst rated aspects on the Likert scale of the first part of the questionnaire and that would need to be improved;With regard to consultation time, an aspect that provides information on the patient’s perception of the time spent in the consultation room, 87.2% of those surveyed considered that they had spent the necessary time. In comparison, 10.4% and 2.4% stated that this time had been shorter and longer than necessary, respectively, which is subjective, as patients are sometimes unaware of the exact time needed to establish an adequate diagnosis and treatment for their pathology. Those patients who considered that the time they spent was less time than they considered necessary were dissatisfied, as they related this to poorer care that could lead to a deterioration in their state of health.

It was found that overall satisfaction was mainly linked to staff qualities in their relationship with the patient (“staff interest in solving problems and delivering what they promise”, “trust that the staff transmits”, “information provided by the doctor” or “personalised treatment”). These variables were similar to those found in the study by Jorge-Cerrudo et al. [45].

On the other hand, we found those aspects that were the most determinant of overall patient satisfaction, showing that the predictor variables of satisfaction were those linked to the interpersonal doctor–patient relationship, the technical–scientific component, accessibility and the comfort of the environment. In order of association: “the technology of the medical equipment”; “the directions to the surgery”; “the confidence that the staff transmits”; “the state of the surgery”; and “the staff’s interest in solving problems”. These results are similar to those obtained by García-Aparicio et al. [28].

After analysing patients’ responses according to their sociodemographic characteristics, statistically significant differences were observed for gender, educational attainment and employment status, with higher satisfaction found in male patients [49], with a level of education up to primary school and no work activity, which reflected the influence of certain sociodemographic factors on patient satisfaction. Therefore, patient satisfaction is not only influenced by their expectations and aspects of the care received, but also by their personal characteristics. Thus, certain sociodemographic characteristics may condition patients’ expectations and, consequently, their satisfaction. The study of the relationship between sociodemographic factors and satisfaction allows us to obtain a more realistic vision of patient subgroups and to adapt those aspects considered to be deficient to the former [24]. However, other studies that analyse the influence of sociodemographic variables on perceived quality present contradictory results, arguing the need for change in the behaviour of healthcare personnel, assuming certain sensitivities. However, making positive discrimination in health care against certain sociodemographic groups just because they belong to them raises ethical doubts. In order to improve the patient’s perception of what he/she expected, informative measures could be used to modify “what is expected” [45].

In addition, the relationship between the overall level of satisfaction expressed by the patients surveyed and those aspects corresponding to the additional questions in the second part of the questionnaire was explored. Statistically significant differences were observed in the variables recommendation of the service received, consultation time and information received about what was happening to them, with higher satisfaction being found in those patients who recommended the service received at the consultation without hesitation; those who considered the consultation time to be as long as necessary or longer than necessary; and those who received sufficient information about what was happening to them.

Finally, the patients’ sociodemographic variables were assessed, related to each aspect of care in the first block of responses (19 items), both of which were recoded. Statistically significant differences were found in the demographic variable age with respect to the item relating to objective quality “staff competence”, where patients aged 66 years and over scored this item higher than the rest of the age ranges. In the demographic variable sex, statistically significant differences were found, mainly with respect to the items relating to subjective quality: “the interest of the staff in fulfilling their promises and solving problems”; “the willingness of the staff to help when needed”; “the confidence that the staff transmits”; “the preparation of the staff to carry out their work”; “the personalised treatment received”; “the ability of the staff to understand the needs”; and “the interest of the nursing staff in the patient”. In all of them, men scored these items higher than women, in contrast to other studies [26], answering that they were more satisfied than expected. With regard to the demographic variable “completed studies”, statistically significant differences were observed, especially with respect to the items relating to objective quality: “the technology of the medical equipment”; “the directions for getting to the consultation”; “the state of the consultation”; “the ease of getting to the hospital or outpatient clinic”; “the interest of the staff in solving problems”; and “the punctuality of the consultation”. In all of them, patients who had completed primary education scored these items higher than those who had completed higher education, and were more satisfied than expected. Finally, in the demographic variable work situation, statistically significant differences were observed with respect to the items relating to objective quality (“the state of the consultation room” and “the information the doctor provides”) and those relating to subjective quality (“the interest of the staff in fulfilling what they promise”, “the interest of the staff in solving problems” and “the preparation of the staff to carry out their work”), with better scores for patients who were not working than for those who were working at the time of the survey.

Moreover, the higher scores observed in the items described above, given by men, patients with up to primary education and those who were not working, coincide with the results obtained previously when analysing the association between the level of satisfaction and the sociodemographic variables, with higher satisfaction being found in these three categories.

On the other hand, it was also observed that men rated more positively those items relating to subjective quality than women, which indicates that, in the latter, expectations are higher in relation to the care and treatment of the staff, while a higher score on the items indicating objective quality was found in patients with primary education, which indicates that those patients with higher education are more demanding in relation to structural aspects and accessibility.

At the end of the questionnaire, there was a section to write opinions or suggestions about some aspects that had not been previously collected in the survey. In general, the comments collected were of thanks and positive opinions for the healthcare staff and the service in general and some suggestions for improvement related to the reduction in waiting times for medical attention and the results of diagnostic tests, as well as those referring to the state of the facilities and furniture.

There are some limitations to this study: The use of questionnaires to assess the level of satisfaction depends on the patient’s expectations. Thus, if these are low, their perceived quality of care will be better (subjectivity bias). The knowledge of the healthcare staff that this study was being carried out may modify their behaviour towards patients and relatives when they feel evaluated (complacency bias). The use of satisfaction surveys may give rise to confounding biases due to the wrong choice of some items that are similar and selection biases, as people with cognitive deficits do not answer the questionnaire. Some participants in satisfaction surveys may give positive and high scores that are not truthful in order to generate socially acceptable responses (complacency bias). Patients who are more likely to complete surveys tend to belong to the two opposite poles of satisfaction (selection bias).

## 5. Conclusions

Based on the results obtained and discussed above, it is possible to conclude that patients had a high degree of satisfaction. The majority of the most highly rated aspects are related to humane treatment and the relationship with the healthcare staff. Among the points for improvement, those related to infrastructure, technology and signposting of the centre or consultation stand out. The study makes it possible to detect areas for improvement in the healthcare provided by the Dermatology Service. In those areas with the lowest ratings, action should be taken to increase the levels of perceived quality, which would lead to both an increase in the degree of patient satisfaction and an improvement in the quality of care.

Possible proposals for improvement include:Better identification of healthcare staff, mainly nurses, but also doctors;Reduction in waiting times for health care and diagnostic test results;Improving the facilities, technologies, infrastructure and signposting of the centres;Providing more information to patients’ relatives.

The use of the same questionnaire in the future will make it possible to evaluate its impact on healthcare after taking appropriate measures to improve the care provided. This ensures, on the one hand, the citizen’s participation in decision making and, on the other hand, generates a feedback process between the patient and the service in order to carry out the continuous improvement of services.

It is clear that satisfaction is closely linked to the qualities and skills of the staff in their relationship with the patient.

## Figures and Tables

**Table 1 healthcare-10-01560-t001:** Perceived quality of health care in the 19 items of the first block of the survey.

	1 Much Worse than I Expected *n* (%)	2 Worse than I Expected *n* (%)	3 As I Expected *n* (%)	4 Better than I Expected *n* (%)	5 Much Better than I Expected *n* (%)	Mean ± SD
Medical equipment technology	9 (3.6)	12 (4.8)	105 (42)	47 (18.8)	77 (30.8)	3.68 ± 1.072
The appearance of the staff	1 (0.4)	0 (0)	58 (23.2)	52 (20.8)	139 (55.6)	4.31 ± 0.854
Directions to the consultation	2 (0.8)	7 (2.8)	78 (31.2)	49 (19.6)	114 (45.6)	4.06 ± 0.976
Staff interest in delivering on their promises	4 (1.6)	11 (4.4)	52 (20.8)	55 (22.0)	128 (51.2)	4.17 ± 1.008
The status of the consultation	3 (1.2)	15 (6.0)	73 (29.2)	43 (17.2)	116 (46.4)	4.02 ± 1.053
The information the doctor gives you	5 (2.0)	14 (5.6)	47 (18.8)	56 (22.4)	128 (51.2)	4.15 ± 1.042
Waiting time to be attended	5 (2.0)	9 (3.6)	44 (17.6)	50 (20.0)	142 (56.8)	4.26 ± 1.002
Ease of getting to the hospital or outpatient clinic	3 (1.2)	1 (0.4)	92 (36.8)	43 (17.2)	111 (44.4)	4.03 ± 0.969
Staff interest in solving problems	7 (2.8)	7 (2.8)	58 (23.2)	46 (18.4)	132 (52.8)	4.16 ± 1.051
Timeliness of the consultation	4 (1.6)	12 (4.8)	46 (18.4)	42 (16.8)	146 (58.4)	4.26 ± 1.021
The speed with which you get what you need or ask for	6 (2.4)	18 (7.2)	54 (21.6)	60 (24.0)	112 (44.8)	4.02 ± 1.083
The willingness of the staff to help you when you need it	2 (0.8)	5 (2.0)	56 (22.4)	52 (20.8)	135 (54.0)	4.25 ± 0.925
The confidence that the staff conveys	6 (2.4)	5 (2.0)	44 (17.6)	57 (22.8)	138 (55.2)	4.26 ± 0.979
The friendliness of the staff in their dealings with the patient	1 (0.4)	3 (1.2)	33 (13.2)	61 (24.4)	152 (60.8)	4.44 ± 0.796
Preparing staff to do their job	3 (1.2)	6 (2.4)	52 (20.8)	47 (18.8)	142 (56.8)	4.28 ± 0.953
The personalised treatment he has received	3 (1.2)	8 (3.2)	51 (20.4)	46 (18.4)	142 (56.8)	4.26 ± 0.971
The capacity of staff to understand the needs	5 (2.0)	7 (2.8)	57 (22.8)	56 (22.4)	125 (50.0)	4.16 ± 1.000
The information the doctor gives to relatives	2 (2.4)	3 (3.6)	21 (25.3)	24 (28.9)	33 (39.8)	4.00 ± 1.012
Nursing staff’s interest in the patient	2 (0.8)	5 (2.0)	59 (23.6)	55 (22.0)	129 (51.6)	4.22 ± 0.928

**Table 2 healthcare-10-01560-t002:** Significant variables in the multivariate study.

	*p*-Value	CI (95%)
Medical equipment technology	0.001	0.304/1.202
Directions to the practice	0.005	−1.301/−0.225
The status of the consultation	0.020	0.097/1.155
Staff interest in solving problems	0.048	0.005/1.399
The confidence that the staff conveys	0.016	0.194/1.911

**Table 3 healthcare-10-01560-t003:** Analysis of the influence of recoded sociodemographic data on overall satisfaction.

		Satisfied (%)	Unsatisfied (%)	*X*^2^*p*-Value
**Gender**	Men	102 (97.1%)	3 (2.9%)	**0.024**
	Women	130 (89.7%)	15 (16.7%)	
Age	≤45 years	116 (92.8%)	9 (7.2%)	0.451
	46–65 years	80 (90.9%)	8 (9.1%)	
	≥66 years	36 (97.3%)	1 (2.7%)	
Marital status	Married	101 (92.7%)	8 (7.3%)	0.940
	Single	131 (92.9%)	10 (7.1%)	
**Completed studies**	Up to Primary	78 (97.5%)	2 (2.5%)	**0.049**
	Higher education	154 (90.6%)	18 (9.4%)	
**Employment status**	No work activity	113 (97.4%)	3 (2.6%)	**0.009**
	Working	119 (88.8%)	15 (11.2%)	

**Table 4 healthcare-10-01560-t004:** Analysis of the influence of different variables on overall satisfaction.

		Satisfied (%)	Unsatisfied (%)	*X*^2^*p*-Value
**Recommendation of the service received**	Without hesitation	219 (99.5%)	1 (0.5%)	**<0.001**
	I have doubts	13 (46.4%)	15 (53.6%	
	Never	0 (0.0%)	2 (100.0%)	
**Consultation time**	What is necessary	209 (95.9%)	9 (4.1%)	**<0.001**
	Less than necessary	17 (65.4%)	9 (34.6%)	
	More than necessary	6 (100.0%)	0 (0.0%)	
Tests or interventions without permission	Yes	8 (100.0%)	0 (0.0%)	0.423
	No	224 (92.6%)	18 (7.4%)	
Know the name of the doctor	Yes	104 (95.4%)	5 (4.6%)	0.160
	No	128 (90.8%)	13 (9.2%)	
Know the name of the nurse	Yes	34 (100.0%)	0 (0.0%)	0.081
	No	198 (91.7%)	18 (8.3%)	
**Sufficient information received**	Yes	205 (99.0%)	2 (1.0%)	**<0.001**
	No	27 (62.8%)	16 (37.2%)	

**Table 5 healthcare-10-01560-t005:** Analysis of the influence of the 19 recoded items with the recoded demographic variables.

Recoded Items (Answer 1 to 3)	Gender	Age	Marital Status	Studies	Labour
The technology of medical equipment	0.061	0.442	0.677	**0.000**	0.678
The appearance of the staff	0.116	**0.038**	0.150	0.061	0.257
Directions to the clinic	0.099	0.237	0.106	**0.007**	0.270
The interest of the staff to deliver what they promise	**0.003**	0.592	0.178	0.100	**0.004**
The condition of the practice	**0.039**	0.092	0.096	**0.020**	**0.034**
The information the doctor gives you	**0.036**	0.596	0.201	0.281	**0.004**
Waiting time to be seen	0.073	0.270	0.640	0.153	0.647
How easy it is to get to the hospital or outpatient clinic	0.223	0.143	0.581	**0.004**	0.585
The interest of the staff in solving problems	**0.002**	0.325	**0.040**	**0.044**	**0.010**
The punctuality of the consultation	0.122	0.380	0.866	**0.048**	0.687
The speed with which you get what you need or ask for	0.146	0.814	0.657	**0.016**	0.352
The willingness of the staff to help you when you need it	**0.049**	0.155	0.210	0.063	0.168
The confidence that the staff gives you	**0.017**	0.979	0.390	0.165	0.065
The friendliness of the staff in their dealings with the patient	0.530	0.311	0.159	0.773	0.444
The preparation of the staff to carry out their work	**0.034**	0.262	0.248	0.219	**0.045**
The personalised treatment you have received	**0.022**	0.633	0.717	0.205	0.091
The ability of the staff to understand your needs	**0.007**	0.833	0.568	0.436	0.098
The information given by the doctor to the relatives	0.592	0.356	**0.024**	0.467	0.405
The interest of the nursing staff in the patient	**0.016**	0.512	0.929	0.220	0.546

## Data Availability

The data presented in this study are available on request from the corresponding author. The data are not publicly available for reasons of the Hospital’s internal data protection regulations.

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
