# Peer review of "Evaluation of the Satisfaction of Patients Seen in the Dermatology Department of a Spanish Tertiary Hospital"

_healthcare, 2022, doi:10.3390/healthcare10081560_

Round 1

Reviewer 1 Report

Dear editors,

Thanks for the opportunity to review this paper. This article describes an evaluation of the satisfaction of patients seen in the outpatient department of the dermatology department of a Spanish tertiary hospital. I believe the article is interesting and can be helpful for other researchers. However, some aspects perhaps could be revised to improve the overall quality. These are the following:

- The authors describe, «The study population consisted of patients attending the outpatient clinics of the CAULE Dermatology Department. The sample consisted of 250 surveys completed by patients and handed out at the exit of the outpatient dermatology department of the CAULE«. In this research, the potential selection bias is paramount, so it would be interesting if the authors could perform three tasks related to this aspect: first, to describe the main features of the eligible population and, of course, the sample obtained. Second, to better describe the recruitment process, to allow the readers to understand the potential bias.

For example, the authors write, «The study included all patients of any age who attended the outpatient dermatology department and who voluntarily signed the informed consent form to participate in the study». However, patients under 16 may not be prepared to answer the survey. Thus, I would invite the authors to describe the inclusion and exclusion criteria better.

Finally, I would propose the authors assess, if possible, this potential bias regarding the external validity of their conclusions.

It would be interesting if the authors described if the survey needed translation or transcultural adaptation.

In the limitations, I would invite the authors to describe the potential selection bias and, if possible, assess it.

In the conclusions section, please remove the sentence «This section is not mandatory but can be added to the manuscript if the discussion is unusually long or complex» :)

I would also propose that the authors remove results from the conclusions section. For example, «The final average score of 4.17», or «statistically significant differences were obtained, with a higher degree of satisfaction being found in those patients who recommended the service received in the consultations without hesitation, those who considered that the consultation time was necessary or longer than necessary, and those who responded that they received sufficient information about what was happening to them». These results are previously described and discussed. Thus, this section would be more precise if only the main conclusions were shown.

Reviewer 2 Report

In the current study, the authors modified SERVQHOS healthcare quality survey model to determine the degree of satisfaction of patients seen in the outpatient department, aiming to improve and offer better quality care. The survey included a large number of participants and has wide coverage. The originality of the study is high, and this satisfaction survey in valuable for the evaluating. I just have a minor question: the current survey was done in dermatology department, did the authors do the same satisfaction survey in other departments? What is their prediction of the survey from other departments?
